# All $4 \times 4$ solutions of the quantum Yang–Baxter equation

**Marius de Leeuw and Vera Posch**

*Hamilton Mathematics Institute*
*School of Mathematics*
*Trinity College Dublin, Dublin, Ireland*

`{deleeuwm,poschv}@tcd.ie`

## Abstract

In this paper, we complete the classification of $4 \times 4$ solutions of the Yang–Baxter equation. Regular solutions were recently classified and in this paper we find the remaining non-regular solutions. We present several new solutions, then consider regular and non-regular Lax operators and study their relation to the quantum Yang–Baxter equation. We show that for regular solutions there is a correspondence, which is lost in the non-regular case. In particular, we find non-regular Lax operators whose $R$-matrix from the fundamental commutation relations is regular but does not satisfy the Yang-Baxter equation. These $R$-matrices satisfy a modified Yang–Baxter equation instead.

# 1 Introduction

In this paper, we solve the Yang Baxter Equation (YBE). This equation was named by Faddeev and Takhtajan after Yang and Baxter [1]. Yang and Baxter encountered this equation in different contexts. Yang studied a one-dimensional many-body quantum mechanical scattering [2] and Baxter consider the eight vertex model in statistical physics [3]. In fact, this equation arises in many different settings in various fields in physics and mathematics. For a brief (historical) overview into the different fields in which Yang-Baxter equation arises, see for instance [4] or the book by Jimbo [5].

Today, the YBE is central to quantum integrability and more interesting and relevant to mathematical physics and applications than ever. This paper concerns itself with solving the YBE and classifying its solutions. In general, this is a very daunting challenge. The Yang–Baxter equation is a matrix equation on an object called the $R$-matrix and takes the form

$$R_{12}(u,v)R_{13}(u,w)R_{23}(v,w) = R_{23}(v,w)R_{13}(u,w)R_{12}(u,v) \tag{1}$$

with $R : V \otimes V \to V \otimes V$, $R_{12} = R \otimes \mathbf{1}$ and $u,v \in \mathbf{C}$. The entries of the $R$-matrix are functions depending on spectral parameters. Hence, the Yang-Baxter equation describes a coupled set of cubic functional equations. Solving this is clearly not a simple task. The Hilbert space $V$ is usually chosen to be $\mathbf{C}^n$ and in this paper we will restrict to $\mathbf{C}^2$.

When the local Hilbert space is two-dimensional, classification of constant solutions to Baxter's equations was initiated in [6] using algebraic geometry. A special case of the Yang–Baxter equation (1) is where all spectral parameters are taken to be equal $u = v = w$. These correspond to so-called constant solutions. In the case of $V = \mathbf{C}^2$ all constant solutions were classified and listed in [7]. The best known solution is the permutation matrix $P$, which acts as a permutation of vector spaces in a tensor product. The other constant solutions have only recently been picked up again by [8], where they discuss interesting algebraic properties described by these matrices.

In general, we are interested in non-constant solutions and differentiable solutions. To this end, we consider $R$-matrices that are analytic in the spectral parameters $u$ and $v$. We consider all the entries of the $R$-matrices to be holomorphic functions, *i.e.* they admit a Taylor expansion

$$R(u,v) = R^{(0)} + uR^{(1)} + vR^{(2)} + uvR^{(3)} + u^2R^{(4)} + \dots \tag{2}$$

Due to the fact that the Yang–Baxter equation is not sensitive to the normalisation of $R$, we can always do this as long as the entries are meromorphic functions. In principle, we can solve the Yang–Baxter equation order by order in the spectral parameters. At zeroth order, we find the constant Yang–Baxter equation. Therefore $R^{(0)}$ needs to be one of the constant solutions found in [7], which we list in Appendix A for completeness.

The most studied set of solutions are the so-called regular solutions where $R^{(0)} = P$. These correspond to spin chains with nearest neighbour interactions and usually correspond to $R$-matrices that appear as scattering matrices in integrable quantum field theories. Traditionally these types of solutions have been studied using symmetry arguments [9, 10] or more direct approaches [11] . More recently an approach based on the boost operator was set-up [12, 13, 14], which resulted in the full classification for $4 \times 4$ regular solutions of the Yang–Baxter equation [15].

So, in order to complete the full classification of $4 \times 4$ solutions we need to consider the remaining constant cases which give non-regular $R$-matrices. In this paper we discuss how these can be extended to non-constant solutions. A first attempt at this was put forward in [16] using a new algorithmic approach to construct solutions . In this paper we are completing the classification $4 \times 4$ non-regular solutions of the quantum Yang–Baxter equation. In section 3 all the models have been listed. Our methods are explained and illustrated by an example in Section 2.

The existence of a quantum integrable model is normally understood as the existence of a tower of commuting conserved charges. These are generated by a Lax operator $L$ which solves the fundamental commutation relations [17, 18]

$$R_{12}(u,v)L_{13}(u)L_{23}(v) = L_{23}(v)L_{13}(u)R_{12}(u,v) \tag{3}$$

By identifying the Lax operator with the $R$-matrix via $L(u) \sim R(u,0)$ this can be seen as a special case of the YBE. Indeed we show that for regular solutions this identification is always possible, so that any integrable model described by a regular Lax operator corresponds to a solution of the Yang–Baxter equation.

However, as we discuss in Section 4, this breaks down the non-regular setting. For an integrable model described by a non-regular Lax operator, we find that the fundamental commutation relations can be solved by $R$-matrices that do *not* satisfy the Yang-Baxter equation. Instead they satisfy a modified Yang Baxter equation We discuss two explicit examples to illustrate this.

## 2  Method

In this section, we will explain the framework that we use to lift a constant solution of the Yang–Baxter equation to a solution that depends on spectral parameters. We assume that the spectral parameters are complex numbers, and we will also assume that the entries of the $R$-matrix admits a Laurent series expansion in terms of the spectral parameters. Since we can identify the solution up to overall normalization this means that we can assume that $R$ is analytic and admits a Taylor expansion. This condition is also necessary from a physical point of view, where one would define the conserved charges of the corresponding integrable models as logarithmic derivatives of the transfer matrix.

### 2.1  Framework

The key to our derivation is to expand around the known constant solutions. Let us show that at the point of coinciding spectral parameters, the $R$-matrix reduces to one of the constant solutions. This fixes the form of the expansion.

**Proposition 1.** *Let $R(u,v)$ be a analytic solution of the Yang-Baxter equation. Then, up to a basis transformation we can write*

$$R(u,v) = R^{(0)}(u_+) + u_- R^{(1)}(u_+) + \frac{u_-^2}{2} R^{(2)}(u_+) + \ldots, \tag{4}$$

*where $u_- = u - v$ and $u_+ = \frac{u+v}{2}$ and $R^{(0)}(u_+)$ is one of the $R$-matrices from [7] where we replace all constants with analytic functions $a \mapsto a(u_+)$.*

*Proof.* Since $R$ is analytic in $u, v$, it will also be analytic in the variable $u_\pm$. In, particular it admits a power series of the form (4). Let us now consider the Yang-Baxter equation and set all the spectral parameters equal. We find

$$R_{12}(u,u)R_{13}(u,u)R_{23}(u,u) = R_{23}(u,u)R_{13}(u,u)R_{12}(u,u). \tag{5}$$

This is exactly the constant Yang-Baxter equation whose solutions where classified in [7]. However, this Yang-Baxter equation is satisfied by $R(u_+)$ and hence all constants that appear in [7] need to be promoted to functions. □

By expanding the Yang-Baxter equation around the points where two of the spectral parameters coincide, we can then derive the following necessary conditions that $R$ needs to satisfy:

$$\begin{aligned} R_{12}^{(0)}(u_+)R_{13}(u,v)R_{23}(u,v) &= R_{23}(u,v)R_{13}(u,v)R_{12}^{(0)}(u_+) \\ R_{12}(u,v)R_{13}^{(0)}(u_+)R_{23}(v,u) &= R_{23}(v,u)R_{13}^{(0)}(u_+)R_{12}(u,v) \\ R_{12}(u,v)R_{13}(u,v)R_{23}^{(0)}(u_+) &= R_{23}^{(0)}(u_+)R_{13}(u,v)R_{12}(u,v) \end{aligned} \tag{6}$$

Plugging (4) into (6) and letting $u \to v$ we can recursively solve for every order of $u - v$ independently. In particular, at leading order we find the constant Yang–Baxter equation again. For the next order we find the following equations

$$R_{12}^{(0)} R_{13}^{(0)} R_{23}^{(1)} + R_{12}^{(0)} R_{13}^{(1)} R_{23}^{(0)} = R_{23}^{(0)} R_{13}^{(1)} R_{12}^{(0)} + R_{23}^{(1)} R_{13}^{(0)} R_{12}^{(0)}, \tag{7}$$

$$R_{12}^{(1)} R_{13}^{(0)} R_{23}^{(0)} - R_{12}^{(0)} R_{13}^{(0)} R_{23}^{(1)} = R_{23}^{(0)} R_{13}^{(0)} R_{12}^{(1)} - R_{23}^{(1)} R_{13}^{(0)} R_{12}^{(0)}, \tag{8}$$

$$R_{12}^{(1)} R_{13}^{(0)} R_{23}^{(0)} + R_{12}^{(0)} R_{13}^{(1)} R_{23}^{(0)} = R_{23}^{(0)} R_{13}^{(1)} R_{12}^{(0)} + R_{23}^{(0)} R_{13}^{(0)} R_{12}^{(1)} \tag{9}$$

where we have omitted the explicit spectral dependence since all the matrices $R_{ab}^{(i)}$ depend on the same spectral parameter. These equations are not all independent as the first two add up to give

the third. Notice that for regular $R$-matrices these linear equations are automatically satisfied, but for non-regular $R$-matrices these give non-trivial constraints. We can also see that $R^{(1)} = R^{(0)}$ is automatically a solution. This degree of freedom corresponds to an overall rescaling of our $R$-matrix and hence we can set the coefficient proportional to $R^{(0)}$ to 0.

For example, if we take $R^{(0)} = 1$, then these equations give that

$$[R^{(1)}_{ij}, R^{(1)}_{kl}] = 0, \tag{10}$$

for all $i, j, k, l$. This basically implies that $R^{(1)}$ needs to be diagonal. Expanding further we get a coupled set of equations that we can solve perturbatively for each choice of $R^{(0)}$. This gives us a set of necessary conditions on the coefficients from the expansion of the $R$-matrix. What then remains is substituting the solution to the above constraints into the Yang–Baxter equation and solve for the remaining degrees of freedom.

## 2.2 Examples

In order to illustrate our method, let us work out two examples to show the different steps that are needed. We will start from two constant solutions as our starting point, namely $R^H$ and $R^A$.

**Example $R^H$** We take the expansion equation (4) and then solve equations (7)-(9) for the elements of the unknown matrix $R^{(1)}$. The components of $R^{(1)}$ are functions $f_i^{(1)}(u+v)$ that depend on the sum of the spectral parameters. Equation (7) then gives rise to a set of linear equations in the functions $f_i^{(1)}$ which can be solved. In this case, we find a solution with a single degree of freedom, which we call $f^{(1)}$ and we get

$$R^{H,(1)} = f^{(1)} \begin{pmatrix} \cdot & \cdot & \cdot & 1 \\ \cdot & \cdot & 1\cdot & \cdot \\ \cdot & 1 & \cdot & \cdot \\ -1 & \cdot & \cdot & \cdot \end{pmatrix} \tag{11}$$

Plugging this back in our expansion(4) we can then expand (6) to second order which leads to a system of equations linear in the components of $R^{H,(2)}$ and quadratic in $R^{H,(1)}$. The elements of $R^{H,(2)}$ are expressed in terms of $f^{(1)}$. To be more explicit we find:

$$R^{H,(2)} = \frac{1}{2}(f^{(1)})^2 \begin{pmatrix} \cdot & \cdot & \cdot & 1 \\ \cdot & \cdot & 1\cdot & \cdot \\ \cdot & 1 & \cdot & \cdot \\ -1 & \cdot & \cdot & \cdot \end{pmatrix} \tag{12}$$

Going to the next higher orders this is a pattern that repeats. $R^{H,(n)}$ will be of the same structure as the previous ones, and it will depend on the elements of the previous $R^{H,(i)}$ $(i < n)$.

At this point, we can recursively show that the corresponding solution to the Yang–Baxter equation should take the form

$$\mathcal{R}^H(u,v) = R^H + (f(u,v) - 1) \begin{pmatrix} \cdot & \cdot & \cdot & 1 \\ \cdot & \cdot & 1\cdot & \cdot \\ \cdot & 1 & \cdot & \cdot \\ -1 & \cdot & \cdot & \cdot \end{pmatrix}. \tag{13}$$

Plugging it into the Yang–Baxter equation we find that it is solvable if and only if

$$f(u,w) = f(u,v)f(v,w). \tag{14}$$

This can be solve by expanding (14) around $v = u$ and yields that

$$f(u,v) \equiv e^{F(u)-F(v)}, \tag{15}$$

for some holomorphic function $F$. We can then use reparameterization invariance to simply define our spectral parameters as $F(u) \to u$ and we find our final answer:

$$\mathcal{R}(u,v) = \begin{pmatrix} 1 & \cdot & \cdot & e^{u-v} \\ \cdot & 1 & e^{u-v}\cdot & \cdot \\ \cdot & e^{u-v} & 1 & \cdot \\ -e^{u-v} & \cdot & \cdot & 1 \end{pmatrix},$$
(16)

which coincides with $\mathcal{R}^G$ from our classification below. This $R$-matrix is of difference form and was listed in [16].

**Example $R^A$**  Let us now illustrate a case which is less straightforward by considering $R^A$. We start by considering (7), but since $R^A$ depends on three free functions $p, q, s$, we find that we need to distinguish several cases. For instance the space of solutions for $R^{(1)}$ will be different for some special values, such as *e.g.* $p = 0 = q$. These different cases will lead to different models that have to be studied separately.

In the remainder of the example, let us continue by selecting the case where $p = q = 1$ and $s = -1$. This leads again to a unique solution to (7) given by

$$R^{A,(1)}(u_+) = \begin{pmatrix} \cdot & \cdot & \cdot & f_1^{(1)}(u_+) \\ \cdot & f_2^{(1)}(u_+) & f_3^{(1)}(u_+) & \cdot \\ \cdot & f_4^{(1)}(u_+) & f_5^{(1)}(u_+) & \cdot \\ f_6^{(1)}(u_+) & \cdot & \cdot & f_7^{(1)}(u_+) \end{pmatrix}.$$
(17)

Plugging this back into our expansion and solving for the second order $R^{(2)}$ we find that

$$R^{A,(2)}(u_+) = \begin{pmatrix} \cdot & \cdot & \cdot & f_1^{(2)}(u_+) \\ \cdot & f_2^{(2)}(u_+) & f_3^{(2)}(u_+) & \cdot \\ \cdot & f_4^{(2)}(u_+) & f_5^{(2)}(u_+) & \cdot \\ f_6^{(2)}(u_+) & \cdot & \cdot & f_7^{(2)}(u_+) \end{pmatrix}.$$
(18)

However, we also find several quadratic equations for the first order functions $f_i^{(1)}$. These can be solved and we find three different solutions

$$R^{A,(1,1)}(u_+) = \begin{pmatrix} \cdot & \cdot & \cdot & f_1^{(1)}(u_+) \\ \cdot & \cdot & f_2^{(1)}(u_+) & \cdot \\ \cdot & f_2^{(1)}(u_+) & \cdot & \cdot \\ f_4^{(1)}(u_+) & \cdot & \cdot & \cdot \end{pmatrix},$$
(19)

$$R^{A,(1,2)}(u_+) = \begin{pmatrix} \cdot & \cdot & \cdot & f^{(1)}(u_+) \\ \cdot & \cdot & \cdot & \cdot \\ \cdot & \cdot & \cdot & \cdot \\ \cdot & \cdot & \cdot & \cdot \end{pmatrix}, \qquad R^{A,(1,3)}(u_+) = \begin{pmatrix} \cdot & \cdot & \cdot & \cdot \\ \cdot & \cdot & \cdot & \cdot \\ \cdot & \cdot & \cdot & \cdot \\ f^{(1)}(u_+) & \cdot & \cdot & \cdot \end{pmatrix}$$
(20)

We see that $R^{A,(1,3)}$ and $R^{A,(1,2)}$ are related by transposition, so they do not describe independent models. Hence, in this case there are two possible paths that describe independent solution..

Let us continue with $R^{A(1,2)}$. Repeating the above procedure we find that the structure recursively repeats itself and leads to the $R$-matrix $\mathcal{R}^E$ from our classification

$$\mathcal{R}^E(u,v) = \begin{pmatrix} 1 & \cdot & \cdot & f(u,v) \\ \cdot & 1 & \cdot & \cdot \\ \cdot & \cdot & 1 & \cdot \\ \cdot & \cdot & \cdot & -1 \end{pmatrix}.$$
(21)

This is easily checked to be a solution of the Yang–Baxter equation.

# 3 Non-regular solutions of the Yang–Baxter equation

As we saw in the example, most constant solutions only allow for a dependence on the spectral parameters in the overall normalisation factor. These types of models trivially satisfy the Yang–Baxter equation. We now present the remaining solutions to the Yang–Baxter equation and we discuss which constant solutions they follow from. Note that we have used invariance under similarity transformations to present the most compact form.

## 3.1 Rank 4

We first list the invertible solutions.

**Diagonal solution**  There is the obvious diagonal solution to the Yang–Baxter equation

$$\mathcal{R}^A(u,v) = \begin{pmatrix} f_1(u,v) & \cdot & \cdot & \cdot \\ \cdot & f_2(u,v) & \cdot & \cdot \\ \cdot & \cdot & f_3(u,v) & \cdot \\ \cdot & \cdot & \cdot & f_4(u,v) \end{pmatrix}. \tag{22}$$

This model can be found from any of the constant solutions that can be reduced to a diagonal one. This matrix is invertible if all the functions along the diagonal are non-zero. Of course, even in the case where some functions vanish, this still gives a solution of the Yang–Baxter equation, albeit of rank lower than 4.

**XY type solution**  There is a solution which takes the form of a XY type Hamiltonian

$$\mathcal{R}^{B,i,j}(u,v) = f(u,v)\sigma^i \otimes \sigma^i + g(u,v)\sigma^j \otimes \sigma^j, \tag{23}$$

where $i,j = x,y,z$ and $\sigma^i$ are the corresponding Pauli matrices. The case where $i = j$ is trivially equivalent to the diagonal case, which leaves us with 3 independent solutions. There is one further special case in this class

$$\mathcal{R}^C(u,v) = f(u,v)\sigma^z \otimes \sigma^z + g(u,v)\sigma^\pm \otimes \sigma^\pm. \tag{24}$$

The $\pm$ sign flips under transposition and hence this is only one independent solution. The case where we have $i,j = \pm$ also solves the Yang–Baxter equation but it is of lower rank and will be discussed below.

**Upper triangular**  We find three models with an upper-triangular $R$-matrix.

$$\mathcal{R}^D(u,v) = \begin{pmatrix} 1 & \cdot & \cdot & \cdot \\ \cdot & q & e^{u-v}(1-qp) & \cdot \\ \cdot & \cdot & p & \cdot \\ \cdot & \cdot & \cdot & -pq \end{pmatrix}, \qquad \mathcal{R}^{D'}(u,v) = \begin{pmatrix} 1 & \cdot & \cdot & \cdot \\ \cdot & q & e^{u-v}(1-qp) & \cdot \\ \cdot & \cdot & p & \cdot \\ \cdot & \cdot & \cdot & 1 \end{pmatrix}. \tag{25}$$

$$\mathcal{R}^E(u,v) = \begin{pmatrix} 1 & \cdot & \cdot & f(u,v) \\ \cdot & 1 & \cdot & \cdot \\ \cdot & \cdot & 1 & \cdot \\ \cdot & \cdot & \cdot & -1 \end{pmatrix}. \tag{26}$$

**General 8-vertex type**  Then there are finally two additional 8-vertex type models that are of difference form

$$\mathcal{R}^F(u,v) = \begin{pmatrix} 1 & \cdot & \cdot & ke^{u-v} \\ \cdot & -1 & \cdot & \cdot \\ \cdot & 2e^{u-v} & 1 & \cdot \\ \cdot & \cdot & \cdot & 1 \end{pmatrix}, \qquad \mathcal{R}^G(u,v) = \begin{pmatrix} 1 & \cdot & \cdot & e^{u-v} \\ \cdot & -1 & e^{u-v} & \cdot \\ \cdot & e^{u-v} & 1 & \cdot \\ -e^{u-v} & \cdot & \cdot & 1 \end{pmatrix} \tag{27}$$

## 3.2 Rank 3

We find one new solution of rank 3

$$\mathcal{R}^H(u,v) = \begin{pmatrix} \cdot & p & e^{u-v}p & f(u,v) \\ \cdot & \cdot & e^{u-v}k & e^{u-v}q \\ \cdot & k & \cdot & q \\ \cdot & \cdot & \cdot & \cdot \end{pmatrix}. \tag{28}$$

$$\mathcal{R}^I(u,v) = \begin{pmatrix} 1 & \cdot & \cdot & \cdot \\ \cdot & \cdot & \cdot & 1 \\ \cdot & e^{u-v} & \cdot & 1-e^{u-v} \\ \cdot & \cdot & \cdot & 1 \end{pmatrix}, \qquad \mathcal{R}^J(u,v) = \begin{pmatrix} 1 & \cdot & \cdot & \cdot \\ \cdot & \cdot & e^{u-v} & \cdot \\ \cdot & e^{u-v} & \cdot & \cdot \\ \cdot & \cdot & \cdot & \cdot \end{pmatrix}. \tag{29}$$

## 3.3 Rank 2

We find three new solutions of rank 2

$$\mathcal{R}^K(u,v) = \begin{pmatrix} \cdot & \cdot & f_1(u,v) & f_2(u,v) \\ \cdot & \cdot & f_3(u,v) & f_4(u,v) \\ \cdot & \cdot & \cdot & \cdot \\ \cdot & \cdot & \cdot & \cdot \end{pmatrix} \tag{30}$$

$$\mathcal{R}^L(u,v) = \begin{pmatrix} \cdot & f_1(u,v) & \cdot & f_2(u,v) \\ \cdot & \cdot & \cdot & \cdot \\ \cdot & f_3(u,v) & \cdot & f_4(u,v) \\ \cdot & \cdot & \cdot & \cdot \end{pmatrix} \tag{31}$$

$$\mathcal{R}^M(u,v) = \begin{pmatrix} \cdot & f_1(u,v) & \cdot & f_2(u,v) \\ \cdot & \cdot & \cdot & f_3(u,v) \\ \cdot & \cdot & \cdot & \cdot \\ \cdot & \cdot & \cdot & \cdot \end{pmatrix} \tag{32}$$

$$\mathcal{R}^N(u,v) = \begin{pmatrix} \cdot & p-k & \frac{k-p}{k-q}(q+k) & f(u,v) \\ \cdot & \cdot & \cdot & q+k \\ \cdot & \cdot & \cdot & \frac{k+q}{k+p}(p-k) \\ \cdot & \cdot & \cdot & \cdot \end{pmatrix} \tag{33}$$

## 3.4 Rank 1

There is one solution of rank 1

$$\mathcal{R}^O(u,v) = \begin{pmatrix} \cdot & f_1(u,v) & f_2(u,v) & f_3(u,v) \\ \cdot & \cdot & \cdot & \cdot \\ \cdot & \cdot & \cdot & \cdot \\ \cdot & \cdot & \cdot & \cdot \end{pmatrix} \tag{34}$$

We checked that we reproduce all the $R$-matrices from [16] and we have completed their classification.

# 4 R-matrices and Lax operators

The relation between solutions of the Yang–Baxter equation and integrable spin chains follows the path of the algebraic Bethe Ansatz. The key idea is to introduce a Lax operator $L$ which will be used to define a transfer matrix that encodes the family of conserved charges that define the integrable model. The $R$-matrix then describes the algebra between the components of the Lax via the so-called $RLL$-relations or fundamental commutation relations (35).

**Proposition 2.** *Consider a Lax operator $L_{an}(u) : V \otimes V \to V \otimes V$ which satisfies the fundamental commutation relations*

$$R_{ab}(u,v)L_{an}(u)L_{bn}(v) = L_{bn}(v)L_{an}(u)R_{ab}(u,v), \tag{35}$$

*for some invertible matrix $R : V \otimes V \to V \otimes V$. Let us furthermore assume that the entries of $L(u)$ are analytic functions of $u$. Then $L$ describes a spin chain with local Hilbert space $V$ with an infinite tower of charges.*

*Proof.* Construct the transfer matrix

$$t(u) = \text{tr}_a\big[L_{aN}(u)\ldots L_{a1}(u)\big]. \tag{36}$$

From the Fundamental Commutation Relations (35), you can show that

$$[t(u), t(v)] = 0 \tag{37}$$

for all values of $u, v$. Since $L$ is analytic, we have the following expansion for the transfer matrix

$$t(u) = \sum_{n=0}^{\infty} \tilde{Q}_n u^n. \tag{38}$$

Equation (37) then implies that $[\tilde{Q}_m, \tilde{Q}_n] = 0$. $\square$

In order for such a system to be integrable you also need to argue that they are independent. This will of course depend on the explicit form of the Lax operator, but generically that will be the case. Traditionally one defines the different conserved charges via the logarithmic derivatives of the transfer matrix (36). So, let us assume that $\log t$ is well-defined and admits a power series in $u$. In particular, let us define

$$Q_{n+1} \equiv \frac{d^n}{du^n} \log t(u)\Big|_{u=0}, \tag{39}$$

then from (37) it follows that

$$[Q_m, Q_n] = 0. \tag{40}$$

When $L$ is a regular operator, *i.e.* $L(0) = P$, then one can show that the operators $Q_n$ are local operators with interaction range at most $n$. In this case, the Hamiltonian is usually identified with $Q_2$. This is a an operator with nearest-neighbour interactions, given by

$$\mathcal{H} \equiv Q_2 = \sum_i PL'_{i,i+1}(0). \tag{41}$$

In this case we see that the Hamiltonian can be identified with the first non-trivial term in the expansion of the Lax operator. For regular solutions there is a clear connection between the Lax operator and regular solutions of the Yang–Baxter equation.

**Proposition 3.** *Let $L$ be a regular Lax operator which satisfies the fundamental commutation relations. Then the corresponding $R$ matrix is regular as well*

$$R(u, u) = P. \tag{42}$$

*Proof.* By evaluating the RLL relations at $u = v = 0$, we find that

$$R(0, 0) \sim P. \tag{43}$$

Since $R$ is fixed up to an overall normalisation, we can set $R(0, 0) = P$ without loss of generality. Now let us expand the Lax operator and the $R$-matrix as

$$L(u) = P(1 + u\mathcal{H} + \sum_{i=2}^{\infty} u^i \check{L}^{(i)}), \qquad R(u, u) = P(1 + \sum_{i=1}^{\infty} u^i \check{R}^{(i)}). \tag{44}$$

We plug this into the RLL relations and set $v = u$ to find

$$P_{ab}(1 + u\check{R}^{(1)}_{ab} + \ldots)P_{an}(1 + u\mathcal{H}_{an} + \ldots)P_{bn}(1 + u\mathcal{H}_{bn} + \ldots) =$$
$$P_{bn}(1 + u\mathcal{H}_{bn} + \ldots)P_{an}(1 + u\mathcal{H}_{an} + \ldots)P_{ab}(1 + u\check{R}^{(1)}_{ab} + \ldots). \tag{45}$$

At leading order this is satisfied by construction and at the next order we find

$$P_{ab}\check{R}_{ab}^{(1)}P_{an}P_{bn} + P_{ab}P_{an}\mathcal{H}_{an}P_{bn} + P_{ab}P_{an}P_{bn}\mathcal{H}_{bn} =$$
$$P_{bn}P_{an}P_{ab}\check{R}_{ab}^{(1)} + P_{bn}P_{an}\mathcal{H}_{an}P_{ab} + P_{bn}\mathcal{H}_{bn}P_{an}P_{ab}, \tag{46}$$

from which it follows that $\check{R}_{ab}^{(1)} \sim 1$ and hence this coefficient can be put to 0 by choosing again an appropriate normalisation of $R$. One can then recursively show that all orders in $R$ can be chosen to vanish. More specifically, by cancelling the permutation operators in the RLL relations we find

$$(1 + \sum_{i=1}^{\infty} u^i \check{R}_{bn}^{(i)})(1 + u\mathcal{H}_{ab} + \ldots)(1 + u\mathcal{H}_{bn} + \ldots) =$$
$$(1 + u\mathcal{H}_{ab} + \ldots)(1 + u\mathcal{H}_{bn} + \ldots)(1 + \sum_{i=1}^{\infty} u^i \check{R}_{ab}^{(i)}). \tag{47}$$

This yields an equation for $\check{R}^{(i)}$ in terms of the lower order expansion terms of the form

$$\check{R}_{bn}^{(i)} + \sum_{k+l+m=i} \check{R}_{bn}^{(k)}\check{L}_{ab}^{(l)}\check{L}_{bn}^{(m)} = \check{R}_{ab}^{(i)} + \sum_{k+l+m=i} \check{L}_{ab}^{(l)}\check{L}_{bn}^{(m)}\check{R}_{ab}^{(k)}, \tag{48}$$

where the indices in the sum run over positive integers, *i.e.* $l, m > 0$. By the indiction hypothesis we have that $\check{R}^{(k)} = 0$ for $k < i$, from which it follows that $\check{R}_{bn}^{(i)} = \check{R}_{ab}^{(i)}$. This again implies that $\check{R}^{(i)} \sim 1$ and can be set to 0 by an appropriate normalisation. $\qquad\blacksquare$

We can now show that there is a one-to-one correspondence between regular Lax operators of integrable spin chains and regular solutions of the Yang–Baxter equation.

**Theorem 1.** *Any R-matrix which satisfies the RLL relations for a regular Lax operator is a solution of the quantum Yang-Baxter equation*

$$R_{12}(u_1, u_2)R_{13}(u_1, u_3)R_{23}(u_2, u_3) = R_{23}(u_2, u_3)R_{13}(u_1, u_3)R_{12}(u_1, u_2), \tag{49}$$

*and satisfies the braiding unitarity relation*

$$R_{21}(v, u)R_{12}(u, v) = 1. \tag{50}$$

*Conversely, any regular solution $R$ of the quantum Yang–Baxter equation can be interpreted as a regular Lax matrix. More specifically, there is a correspondence between regular solutions of the Yang–Baxter equations and regular Lax operators defined by $L(u) \simeq R(u, 0)$.*

*Proof.* The last part of the theorem is straightforward to prove. From the Yang–Baxter equation it directly follows that $L(u) \simeq R(u, 0)$ satisfies the fundamental commutation relations (35). Conversely, consider the fundamental commutation relations and set $v = 0$. This gives

$$R_{an}(u, 0)L_{ab}(u) = L_{an}(u)R_{ab}(u, 0). \tag{51}$$

Since $L$ is regular, it is invertible and we have that

$$L_{an}^{-1}(u)R_{an}(u, 0) = R_{ab}(u, 0)L_{ab}^{-1}(u). \tag{52}$$

Both sides have a different index structure and act on different spaces. The only solution to this equation is that $L_{an}^{-1}(u)R_{an}(u, 0) \sim 1$, which completes the first part of the proof.

Let us now prove the first part of the theorem. Let $L(u)$ be an invertible Lax operator that satisfies the RLL relations (35). From the fundamental commutation relations relations we find that

$$[R_{21}(v, u)R_{12}(u, v), L_{1n}(u)L_{2n}(v)] = 0. \tag{53}$$

Let us define the object $R_{21}R_{12} \equiv X_{12}$. From Proposition 3 and our assumption that the $R$-matrix is analytic, we see that $X$ admits a power series of the form

$$X(u, v) = 1 + \sum_{i=1}^{\infty} u_-^i X^{(i)}(u_+), \tag{54}$$

where $u_\pm$ are defined as in Proposition 1. Let us define $\check{L} \equiv PL$. We can then, analogously to the proof of the previous lemma, expand this around the point $u = v$, or equivalently $u_- = 0$. At leading order this equation is trivially satisfied. At the next order we find

$$X_{2n}^{(1)}(u)\check{L}_{12}(u)\check{L}_{2n}(u) = \check{L}_{12}(u)\check{L}_{2n}(u)X_{12}^{(1)}(u). \tag{55}$$

Using the expansion of the Lax operator (44), we then find

$$X_{2n}^{(1)}(u)(1 + u\mathcal{H}_{12} + \ldots)(1 + u\mathcal{H}_{2n} + \ldots) = (1 + u\mathcal{H}_{12} + \ldots)(1 + u\mathcal{H}_{2n} + \ldots)X_{12}^{(1)}(u). \tag{56}$$

Expanding this as a power series in $u$ proves that $X^{(1)} \sim 1$. From this one can easily show by induction that $X \sim 1$ and hence that the $R$-matrix satisfies braiding unitarity. Moreover, it is easy to show that the proportionality factor needs to be symmetric under changing $u, v$ and because of this it can be absorbed into the prefactor of the $R$-matrix, making braiding unitarity exact.

What remains to show is that $R$ satisfies the quantum Yang–Baxter equation. To show this, we need to consider a triple product of Lax operators

$$L_{1n}(u)L_{2n}(v)L_{3n}(w). \tag{57}$$

Then you can show that

$$[R_{32}R_{31}R_{21}R_{23}R_{13}R_{12}, L_{1n}L_{2n}L_{3n}] = 0, \tag{58}$$

where we have suppressed the explicit spectral parameter dependence. This is illustrated in Figure 1. One can use the $R$-matrices in the above chain to permute the Lax operators. The chain is chosen such that it leaves $L_1L_2L_3$ invariant.

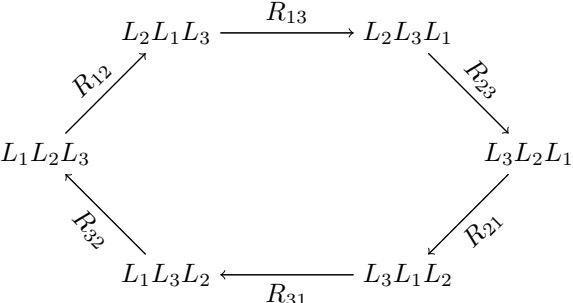

Figure 1: Acting with different $R$-matrices will permute the Lax operators in a the triple product $L_1L_2L_3$ by using the fundamental commutation relations. If we follow the arrows in the diagram, we see that $R_{32}R_{31}R_{21}R_{23}R_{13}R_{12}$ commutes with $L_1L_2L_3$.

For brevity, let us write

$$A_{123} \equiv R_{32}R_{31}R_{21}R_{23}R_{13}R_{12}. \tag{59}$$

We have already shown that for $w = 0$ the $R$-matrix reduces to the Lax operator. We assume that the fundamental commutation relations hold, which this means that the YBE is satisfied in this case

$$R_{12}(u,v)R_{13}(u,0)R_{23}(v,0) = R_{23}(v,0)R_{13}(u,0)R_{12}(u,v). \tag{60}$$

Let us make use of this fact by expanding around the case where one of the spectral parameters vanishes. In particular, consider

$$A_{123}(u,v,w)L_{1n}(u)L_{2n}(v)L_{3n}(w) = L_{1n}(u)L_{2n}(v)L_{3n}(w)A_{123}(u,v,w) \tag{61}$$

We set $u = 0$ and use regularity to get

$$A_{123}(0,v,w)P_{1n}L_{2n}(v)L_{3n}(w) = P_{1n}L_{2n}(v)L_{3n}(w)A_{123}(0,v,w) \tag{62}$$

Cancelling the permutation operators then gives

$$A_{n23}(0, v, w)L_{2n}(v)L_{3n}(w) = L_{2n}(v)L_{3n}(w)A_{123}(0, v, w) \tag{63}$$

We see that the left hand side does not depend on site 1 anymore. Hence $A_{123}(0, v, w) = \tilde{A}_{23}(v, w)$ and we obtain

$$\tilde{A}_{23}(v, w)L_{2n}(v)L_{3n}(w) = L_{2n}(v)L_{3n}(w)\tilde{A}_{23}(v, w). \tag{64}$$

But now we recognise the same equations as in the case of braiding unitarity (53) and we can similarly conclude that $A_{123}(0, v, w) \sim 1$. Let us now expand to the next order in $u$. We get

$$A_{123}(0, v, w)L'_{1n}(0)L_{2n}(v)L_{3n}(w) + A'_{123}(0, v, w)P_{1n}L_{2n}(v)L_{3n}(w) = \tag{65}$$
$$L'_{1n}(0)L_{2n}(v)L_{3n}(w)A_{123}(0, v, w) + P_{1n}L_{2n}(v)L_{3n}(w)A'_{123}(0, v, w). \tag{66}$$

Using $A_{123}(0, v, w) \sim 1$ we derive

$$A'_{123}(0, v, w)P_{1n}L_{2n}(v)L_{3n}(w) = P_{1n}L_{2n}(v)L_{3n}(w)A'_{123}(0, v, w), \tag{67}$$

which is exactly the same equation that we just solved. Hence, by induction we find that $A_{123}(u, v, w) = f(u, v, w)1$, where $f$ is an analytic function.

Concluding, we showed that

$$R_{32}(w, v)R_{31}(w, u)R_{21}(v, u)R_{23}(v, w)R_{13}(u, w)R_{12}(u, v) = f(u, v, w) \tag{68}$$

By braiding unitarity we can derive

$$R_{23}(v, w)R_{13}(u, w)R_{12}(u, v) = f(u, v, w)R_{12}(u, v)R_{13}(u, w)R_{23}(v, w) \tag{69}$$

This is not quite the Yang Baxter equation. We still need to show $f(u, v, w) = 1$. Using Braiding unitarity again (68) also leads to the following rewriting

$$R_{32}(w, v)R_{31}(w, u)R_{21}(v, u) = f(u, v, w)R_{21}(v, u)R_{31}(w, u)R_{32}(w, v) \tag{70}$$
$$R_{32}(w, v)P_{13}R_{13}(w, u)P_{13}R_{21}(v, u) = f(u, v, w)R_{21}(v, u)P_{13}R_{13}(w, u)P_{13}R_{32}(w, v) \tag{71}$$
$$R_{12}(w, v)R_{13}(w, u)R_{23}(v, u) = f(u, v, w)R_{23}(v, u)R_{13}(w, u)R_{12}(w, v) \tag{72}$$

We can now relabel the parameters $u \to w$ and $w \to u$ and obtain

$$R_{12}(u, v)R_{13}(u, w)R_{23}(v, w) = f(w, v, u)R_{23}(v, w)R_{13}(u, w)R_{12}(u, v). \tag{73}$$

By comparing this against (69) we conclude that

$$f(u, v, w) = \frac{1}{f(w, v, u)} \tag{74}$$

Therefore $f$ has to be of the form

$$f(u, v, w) = \pm e^{A(v, w) - A(u, w)} \tag{75}$$

By our identification of $R$ with the Lax operator, we know that putting any of the parameters to zero leads to $f(0, v, w) = f(u, 0, w) = f(u, v, 0) = 1$. Together with (75), this leads to $A = const$ and therefore $f = const = 1$. Which proves that any matrix $R$, satisfying the fundamental commutation relations for a regular Lax operator, is a regular solution of the Yang Baxter Equation.

$\square$

# 5 Non-regular Lax operators and the modified YBE

For non-regular solutions of the Yang-Baxter equation the correspondence between Lax operators and $R$-matrices is lost. As we saw in Proposition 2, the defining relation that implies a tower of conserved charges and integrability are the fundamental commutation relations (35). It is clear that we can

use every non-regular solution of the Yang–Baxter equation that we found as such a Lax operator $L(u) \equiv \mathcal{R}(u,0)$. It is clear that the corresponding $R$-matrix from the fundamental commutation relations exists. However, unlike in the regular case, there might be a more general solution. In order to show this, we now consider

$$\tilde{R}_{12}(u,v)L_{13}(u)L_{23}(v) = L_{23}(v)L_{13}(u)\tilde{R}_{12}(u,v) \tag{76}$$

With $L$ a non-regular Lax matrix as defined above. We can then simply solve for $\tilde{R}$ and see what we find.

For some solutions, there is a more general $\tilde{R}$-matrix than the original $R$-matrix. In particular, we find that all of the $\tilde{R}$ matrices listed below can be chosen to have a regular limit. Furthermore, unlike in the regular case, $\tilde{R}$ *does not solve the YBE*, but solves a modified Yang-Baxter equation instead

$$\tilde{R}_{12}(u,v)\tilde{R}_{13}(u,w)\tilde{R}_{23}(v,w) = M_{123}(u,v,w)\tilde{R}_{23}(v,w)\tilde{R}_{13}(u,w)R_{12}(u,v), \tag{77}$$

or, alternatively,

$$\tilde{R}_{12}(u,v)\tilde{R}_{13}(u,w)\tilde{R}_{23}(v,w) - \tilde{R}_{23}(v,w)\tilde{R}_{13}(u,w)R_{12}(u,v) = \tilde{M}_{123}(u,v,w), \tag{78}$$

The new matrix $\tilde{R}$-matrix does satisfy equation (58), but since the Lax operator is not regular anymore there are more solutions to this equation that just the identity operator. This is the crucial point where the regular and non-regular solutions differ and where the proof of the previous section breaks down.

There will be instances when $\tilde{R}$ satisfies the usual Yang–Baxter equation and is not given by the non-regular $R$-matrix that we started out from. In these cases we can interpret that $R$-matrix as a Lax operator of a different spin chain. In particular, there will instances where the $R$-matrix can for instance be seen as a non-regular Lax operator of the $R$-matrix of the XXX spin chain. This is useful in solving these models via the Algebraic Bethe Ansatz since the known fundamental commutations can be used and the only thing that changes is the action of the monodromy matrix on the reference state.

Certain deformations of the YBE are well-known. For example in [19] a different modified YBE has been proposed. Of course there is also the generalised or twisted YBE [20] [21] and other proposed models such as the dynamical YBE [22]. So far we have not seen a clear correspondence from any of these equations to ours.

## 5.1 Example via Braiding unitarity

An alternative way to see this is to note that a lot of these $R$-matrices do not satisfy braiding unitarity. Because of this, we find that there are two possible solutions of the fundamental commutation relations, namely $R(u,v)$ and $(PR(v,u)P)^{-1} \equiv \hat{R}$. For solutions that satisfy braiding unitarity, these obviously coincide. If they do not, then any linear combination of these will be a solution to the fundamental commutation relations and while both $R(u,v)$ and $(PR(v,u)P)^{-1}$ satisfy the Yang–Baxter equation separately, their sum will generically not. A perfect example of this if $\mathcal{R}^G$. It is easy to show that this model does not satisfy braiding unitarity and we find

$$\hat{\mathcal{R}}^G \sim \begin{pmatrix} 1 & 0 & 0 & -e^{v-u} \\ 0 & 1 & e^{v-u} & 0 \\ 0 & e^{v-u} & -1 & 0 \\ e^{v-u} & 0 & 0 & 1 \end{pmatrix}. \tag{79}$$

Now, we can show by direct computation that the most general solution of the fundamental commutation relations when $L(u) = \mathcal{R}^G(u,0)$ is given by

$$\tilde{\mathcal{R}}^G = f(u,v)\mathcal{R}^G + g(u,v)\hat{\mathcal{R}}^G. \tag{80}$$

It is also easy to check that $\tilde{\mathcal{R}}^G$ does not satisfy the Yang–Baxter equation unless $fg = 0$ or $f = g$. Note that in the latter case $\tilde{\mathcal{R}}^G$ is a regular solution of the Yang–Baxter equation and is of the form 8 vertex A from [13]. In other words, we find that $\mathcal{R}^G$ can be interpreted as a non-regular Lax operator for the $R$-matrix of a free fermion model.

## 5.2 Other example

Apart from (80) we found several more cases of models with a non-trivial $\tilde{\mathcal{R}}$. In some cases we find that $\tilde{\mathcal{R}}$ can be chosen to be regular and satisfying the Yang–Baxter equation. In this case, the non-regular $R$-matrix can be interpreted as a non-regular Lax operator corresponding satisfying the fundamental commutation relations with a known $R$-matrix.

**Model $\mathcal{R}^C$**

$$\tilde{\mathcal{R}}^C(u,v) = \begin{pmatrix} g_1(u,v) & \cdot & \cdot & g_3(u,v) \\ \cdot & g_2(u,v) & \frac{f(u,0)}{f(v,0)}(g_1(u,v)+g_2(u,v)) & \cdot \\ \cdot & \frac{f(v,0)}{f(u,0)}(g_1(u,v)+g_2(u,v)) & g_2(u,v) & \cdot \\ \cdot & \cdot & \cdot & g_1(u,v) \end{pmatrix}, \qquad (81)$$

for free functions $g_1, g_2$. We see that this $\tilde{\mathcal{R}}$ reduces to the permutation operator at $u = v$ if we choose $g_1$ and $g_2$ such that

$$g_1(u,u) = 1, \qquad\qquad g_2(u,u) = 0, \qquad\qquad g_3(u,u) = 0. \qquad (82)$$

This solution $\tilde{\mathcal{R}}$ is regular and satisfies the Yang–Baxter equation exactly when

$$g_1(u,v) = g(u,v), \qquad\qquad g_2(u,v) = \frac{g(u,v)}{G(u) - G(v) - 1}, \qquad (83)$$

where $g$ is an free function and $G' = g$. If we set $g_3 = 0$, then this is simply the $R$-matrix of the twisted XXX spin chain up to a local basis transformation.

# 6 Conclusions and outlook

In this paper we completed the classification of $4 \times 4$ solutions of the quantum Yang–Baxter equation by considering non-regular $R$-matrices. We apply a perturbative approach where we expand about the non-regular solution $R(u,u)$ which were classified previously [7]. We reproduce the results from [16] and extend their results to solutions that are of non-difference form and of lower rank.

We furthermore show that there is a one-to-one relation between regular solutions of the Yang–Baxter equation and regular $R$-matrices. However, for non-regular models this identification is lost. We find for instance that some non-regular $R$-matrices can be interpreted as non-regular Lax operator that satisfy the fundamental commutation relations for a regular $R$-matrix. Generically the $R$-matrices that describe the fundamental commutation relations in these cases satisfy a modified Yang–Baxter equation.

It would be interesting to further understand the modified Yang–Baxter equation and to find if there is a universal form to the modification. So far we have not been able to identify one. Furthermore, extending our results to higher dimensions is another possible avenue of research. Maybe the approach of [8] can used to understand certain classes of non-regular solutions. Finally, it could also be interesting find the physical properties of the models corresponding to our solutions. For instance the spin chains or the quantum circuits that are associated to them.

**Acknowledgements.** We are grateful to S. Shatashvili and L. Takhtajan for important remarks on the introduction. We would like to thank L. Corcoran, V. Grivtsev and A.L. Retore for useful discussions. MdL was supported in part by SFI and the Royal Society for funding under grants UF160578, RGF\ R1\ 181011, RGF\8EA\180167 and RF\ ERE\ 210373. MdL is also supported by ERC-2022-CoG - FAIM 101088193. VP was supported by ERC-2022-CoG - FAIM 101088193.

# A Constant solutions

The starting point for our analysis are the constant solutions of the Yang–Baxter equation. They were classified in [7] and we list them here. We omit the permutation operator, which gives rise to regular solutions who were classified already in [15]. We group the matrices according to their rank.

## A.1 Rank 4

Here we list the invertible $R$-matrices. These were considered recently in [8] as well.

$$R^A = \begin{pmatrix} 1 & \cdot & \cdot & \cdot \\ \cdot & q & \cdot & \cdot \\ \cdot & \cdot & p & \cdot \\ \cdot & \cdot & \cdot & s \end{pmatrix}, \qquad R^B = \begin{pmatrix} 1 & q & p & s \\ \cdot & 1 & \cdot & p \\ \cdot & \cdot & 1 & q \\ \cdot & \cdot & \cdot & 1 \end{pmatrix} \tag{84}$$

$$R^C = \begin{pmatrix} 1 & \cdot & \cdot & \cdot \\ \cdot & q & \cdot & \cdot \\ \cdot & 1-qp & p & \cdot \\ \cdot & \cdot & \cdot & k \end{pmatrix}, \qquad R^D = \begin{pmatrix} \cdot & \cdot & \cdot & q \\ \cdot & \cdot & 1 & \cdot \\ \cdot & 1 & \cdot & \cdot \\ p & \cdot & \cdot & \cdot \end{pmatrix}, \tag{85}$$

$$R^E = \begin{pmatrix} 1 & -p & p & pq \\ \cdot & 1 & \cdot & -q \\ \cdot & \cdot & 1 & q \\ \cdot & \cdot & \cdot & 1 \end{pmatrix}, \qquad R^F = \begin{pmatrix} 1 & \cdot & \cdot & k \\ \cdot & q & \cdot & \cdot \\ \cdot & 1-q & 1 & \cdot \\ \cdot & \cdot & \cdot & -q \end{pmatrix}, \tag{86}$$

$$R^G = \begin{pmatrix} 1+2q-q^2 & \cdot & \cdot & 1-q^2 \\ \cdot & 1+q^2 & 1-q^2 & \cdot \\ \cdot & 1-q^2 & 1+q^2 & \cdot \\ 1-q^2 & \cdot & \cdot & 1-2q-q^2 \end{pmatrix}, \qquad R^H = \begin{pmatrix} 1 & \cdot & \cdot & 1 \\ \cdot & -1 & 1 & \cdot \\ \cdot & 1 & 1 & \cdot \\ -1 & \cdot & \cdot & 1 \end{pmatrix}, \tag{87}$$

$$R^I = \begin{pmatrix} 1 & \cdot & \cdot & 1 \\ \cdot & -1 & \cdot & \cdot \\ \cdot & \cdot & -1 & \cdot \\ \cdot & \cdot & \cdot & 1 \end{pmatrix}. \tag{88}$$

## A.2 Rank 3

$$R^J = \begin{pmatrix} p+q & \cdot & \cdot & \cdot \\ \cdot & q & \cdot & q \\ \cdot & \cdot & p+q & \cdot \\ \cdot & p & \cdot & p \end{pmatrix}, \qquad R^K = \begin{pmatrix} \cdot & p & p & \cdot \\ \cdot & \cdot & k & q \\ \cdot & k & \cdot & q \\ \cdot & \cdot & \cdot & \cdot \end{pmatrix}, \tag{89}$$

$$R^L = \begin{pmatrix} 1 & \cdot & \cdot & \cdot \\ \cdot & \cdot & \cdot & 1 \\ \cdot & 1 & \cdot & \cdot \\ \cdot & \cdot & \cdot & 1 \end{pmatrix}, \qquad R^M = \begin{pmatrix} 1 & \cdot & \cdot & \cdot \\ \cdot & \cdot & 1 & \cdot \\ \cdot & 1 & \cdot & \cdot \\ \cdot & \cdot & \cdot & \cdot \end{pmatrix}. \tag{90}$$

## A.3 Rank 2

$$R^N = \begin{pmatrix} \cdot & (q-k)(p^2-k^2) & (q+k)(p^2-k^2) & s \\ \cdot & \cdot & \cdot & (q^2-k^2)(p+k) \\ \cdot & \cdot & \cdot & (q^2-k^2)(p-k) \\ \cdot & \cdot & \cdot & \cdot \end{pmatrix}, \quad R^O = \begin{pmatrix} 1 & 1 & 1 & \cdot \\ \cdot & \cdot & \cdot & \cdot \\ \cdot & \cdot & \cdot & \cdot \\ \cdot & \cdot & \cdot & 1 \end{pmatrix} \tag{91}$$

$$R^P = \begin{pmatrix} \cdot & p & \cdot & q \\ \cdot & \cdot & \cdot & \cdot \\ \cdot & k & \cdot & \cdot \\ \cdot & \cdot & \cdot & \cdot \end{pmatrix} \qquad R^Q = \begin{pmatrix} \cdot & p & \cdot & \cdot \\ \cdot & \cdot & \cdot & q \\ \cdot & \cdot & \cdot & \cdot \\ \cdot & \cdot & \cdot & \cdot \end{pmatrix}. \tag{92}$$

## A.4 Rank 1

$$R^S = \begin{pmatrix} \cdot & p & q & \cdot \\ \cdot & \cdot & \cdot & \cdot \\ \cdot & \cdot & \cdot & \cdot \\ \cdot & \cdot & \cdot & \cdot \end{pmatrix} \qquad R^T = \begin{pmatrix} \cdot & \cdot & \cdot & \cdot \\ \cdot & p & q & \cdot \\ \cdot & \cdot & \cdot & \cdot \\ \cdot & \cdot & \cdot & \cdot \end{pmatrix} \tag{93}$$

# B   $\tilde{\mathcal{R}}$

The most general form that solves the fundamental commutation relation for the non-regular solutions of the Yang–Baxter equation. We present them for generic values of the parameters.

## B.1   $\tilde{\mathcal{R}}_c$

$$\tilde{\mathcal{R}}_c(u,v) = \begin{pmatrix} g_1(u,v) & \cdot & \cdot & g_3(u,v) \\ \cdot & g_2(u,v) & \frac{f(u,0)}{f(v,0)}(g_1(u,v)+g_2(u,v)) & \cdot \\ \cdot & \frac{f(v,0)}{f(u,0)}(g_1(u,v)+g_2(u,v)) & g_2(u,v) & \cdot \\ \cdot & \cdot & \cdot & g_1(u,v) \end{pmatrix} \tag{94}$$

## B.2   $\tilde{\mathcal{R}}_d$

$$\tilde{\mathcal{R}}_d(u,v) = \begin{pmatrix} g_1(u,v) & \cdot & \cdot & & \cdot \\ \cdot & g_2(u,v) & e^{-u+v}(g_1(u,v) - \frac{g_2(u,v)}{q}) & & \cdot \\ \cdot & g_3(u,v) & \frac{1}{q}(g_1(u,v) - e^{-u+v}g_3(u,v)) & & \cdot \\ \cdot & \cdot & \cdot & -\frac{g_2(u,v)}{q} + e^{-u+v}g_3(u,v) \end{pmatrix} \tag{95}$$

## B.3   $\tilde{\mathcal{R}}_e$

$$\tilde{\mathcal{R}}_e(u,v) = \begin{pmatrix} g_1(u,v) & \cdot & \cdot & g_4(u,v) \\ \cdot & g_2(u,v) & g_3(u,v) & \cdot \\ \cdot & \frac{f(v,0)}{f(u,0)}(g_1(u,v)-g_2(u,v)) & g_1(u,v) - \frac{f(v,0)}{f(u,0)}g_3(u,v) & \cdot \\ \cdot & \cdot & \cdot & -g_2(u,v) + \frac{f(v,0)}{f(u,0)}g_3(u,v) \end{pmatrix} \tag{96}$$

## B.4   $\tilde{\mathcal{R}}_f$

$$\tilde{\mathcal{R}}_f(u,v) = \begin{pmatrix} f_1(u,v) & \cdot & \cdot & f_4(u,v) \\ \cdot & f_2(u,v) & f_3(u,v) & \cdot \\ \cdot & F_1 & F_2 & \cdot \\ F_3 & \cdot & \cdot & F_4 \end{pmatrix} \tag{97}$$

$$F_1 = \frac{2}{k}f_4(u,v) + e^{v-u}(f_1(u,v) + f_2(u,v))$$

$$F_2 = -f_1(u,v) + e^{v-u}(f_3(u,v) + \frac{2}{k}f_4(u,v))$$

$$F_3 = \frac{2}{k}(e^{v-u}(f_1(u,v) + f_2(u,v)) - f_3(u,v))$$

$$F_4 = f_2(u,v) + e^{v-u}(\frac{2}{k}f_4(u,v) + f_3(u,v))$$

$$\tag{98}$$

## B.5   $\tilde{R}_g$

$$\tilde{R}_g(u,v) = \begin{pmatrix} f_1(u,v) & 0 & 0 & f_2(u,v) \\ 0 & H_1(u,v) & H_2(u,v) & 0 \\ 0 & H_2(u,v) & -H_1(u,v) & 0 \\ -f_2(u,v) & 0 & 0 & f_1(u,v) \end{pmatrix} \tag{99}$$

$$H_1(u,v) = \frac{e^{2u}f_1(u,v) - e^{2v}f_1(u,v) - 2e^{u+v}f_2(u,v)}{e^{2u} + e^{2v}}$$

$$H_2(u,v) = \frac{2e^{u+v}f_1(u,v) + e^{2u}f_2(u,v) - e^{2v}f_2(u,v)}{e^{2u} + e^{2v}}$$

$$\tag{100}$$

## B.6 $\tilde{\mathcal{R}}_k$

$$\tilde{\mathcal{R}}_k(u,v) = \begin{pmatrix} g_1(u,v) & g_2(u,v) & g_3(u,v) & G_5(u,v) \\ g_4(u,v) & G_6(u,v) & g_6(u,v) & g_5(u,v) \\ g_7(u,v) & g_8(u,v) & G_7(u,v) & g_9(u,v) \\ G_1(u,v) & G_2(u,v) & G_3(u,v) & G_4(u,v) \end{pmatrix} \tag{101}$$

$$
\begin{aligned}
G_1(u,v) &= -g_1(u,v) - g_2(u,v) - g_7(u,v) \\
G_2(u,v) &= -g_2(u,v) - g_4(u,v) + g_5(u,v) - g_6(u,v) - g_8(u,v) \\
G_3(u,v) &= -g_3(u,v) - g_5(u,v) - g_7(u,v) + g_8(u,v) - g_9(u,v) \\
G_4(u,v) &= g_1(u,v) - g_2(u,v) - g_3(u,v) - g_4(u,v) - g_9(u,v) \\
G_5(u,v) &- g_1(u,v) + g_2(u,v) + g_3(u,v) \\
G_6(u,v) &= g_4(u,v) - g_5(u,v) + g_4(u,v) \\
G_7(u,v) &= g_7(u,v) - g_8(u,v) + g_9(u,v)
\end{aligned}
\tag{102}
$$

## B.7 $\tilde{\mathcal{R}}_i$

$$\tilde{\mathcal{R}}_i(u,v) = \begin{pmatrix} 1 & 0 & 0 & 0 \\ 0 & 0 & g_1(u,v) & 1 - g_1(u,v) \\ 0 & e^{u-v} & 0 & 1 - e^{u-v} \\ 0 & 0 & 0 & 1 \end{pmatrix} \tag{103}$$

## B.8 $\tilde{\mathcal{R}}_j$

$$\tilde{\mathcal{R}}_j(u,v) = \begin{pmatrix} 1 & 0 & 0 & 0 \\ 0 & 0 & e^{u-v} & 0 \\ 0 & e^{u-v} & 0 & 0 \\ 0 & 0 & 0 & g_1(u,v) \end{pmatrix} \tag{104}$$

## B.9 $\tilde{\mathcal{R}}_l$

Any general matrix solves the Lax equation for $\mathcal{R}_l$

## B.10 $\tilde{\mathcal{R}}_m$

$$\tilde{\mathcal{R}}_n = \begin{pmatrix} 1 & g_1(u,v) & g_2(u,v) & g_3(u,v) \\ 0 & g_4(u,v) & g_5(u,v) & g_6(u,v) \\ 0 & \frac{f_1(u,0)f_3(v,0)}{f_1(v,0)f_3(u,0)} & 0 & -\frac{(f_2(v,0)f_3(u,0)-f_2(u,0)f_3(v,0))}{f_1(v,0)f_3(u,0)} \\ 0 & 0 & 0 & 1 \end{pmatrix} \tag{105}$$

## B.11 $\tilde{\mathcal{R}}_m$

$$\tilde{\mathcal{R}}_m = \begin{pmatrix} g_1(u,v) & g_2(u,v) & g_3(u,v) & g_4(u,v) \\ 0 & g_5(u,v) & g_1(u,v) - \frac{(k-p)g_5(u,v)}{k-q} & g_6(u,v) \\ 0 & g_1(u,v) - \frac{(k-p)g_5(u,v)}{k-q} & \frac{(k-p)^2 g_5(u,v)}{(k-q)^2} & F(u,v) \\ 0 & 0 & 0 & g_1(u,v) \end{pmatrix} \tag{106}$$

$$F(u,v) = -\frac{g_1(u,v)(pf_3(u) - pf_3(v) - qf_3(u) + qf_3(v))}{(k-p)(k-q)} + \frac{(k-p)(k+q)g_2(u,v)}{(k+p)(k-q)} - \frac{(-k-q)g_3(u,v)}{k+p} - \frac{(k-p)g_6(u,v)}{k-q} \tag{107}$$

## B.12 $\tilde{\mathcal{R}}_o$

$$\tilde{\mathcal{R}}_o = \begin{pmatrix} 1 & g_1(u,v) & g_2(u,v) & g_3(u,v) \\ 0 & 0 & \frac{f_1(u,0)f_3(v,0)}{f_1(v,0)f_3(u,0)} & -\frac{(f_2(v,0)f_3(u,0)-f_2(u,0)f_3(v,0))}{f_1(v,0)f_3(u,0)} \\ 0 & g_4(u,v) & g_5(u,v) & g_6(u,v) \\ 0 & 0 & 0 & 1 \end{pmatrix} \tag{108}$$

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
