# Peer review of "All 4 x 4 solutions of the quantum Yang-Baxter equation"

_SciPost Physics_

## Round 3 · Referee Report · Anonymous (Referee 1) · 2025-7-27

Strengths

Well written.

Weaknesses

Possible physical applications are not discussed.

Report

The Yang-Baxter equation is the fundamental equation in the theory of quantum integrable models. It is a tantalizing task to classify its solutions and finding new ones is a remarkable achievement. The authors of the manuscript, to the best of my knowledge, obtained new solutions of the Yang-Baxter equation. The new solutions are of the non-regular type and are associated with an interesting modified Yang-Baxter equation. The new results have the potential to motivate further investigations. Therefore I recommend the paper for publication after minor revisions listed below.

Requested changes

  • I think there is a typo in eq. (16): matrix element R[2,2] should be -1 ?

  • I think it is important to cite some standard references in Sec. 4 and clarify what is new and what is already known. Proposition 2, for example, is standard.

  • By the way, I think Proposition 2 could be stated in a more precise form: there is a infinite tower of charges in the thermodynamic limit, but not for a finite chain. The following propositions may be adjusted accordingly.

Recommendation

Ask for minor revision

---

## Round 3 · Referee Report · Anonymous (Referee 2) · 2025-8-1

Strengths

  1. accessible
  2. sets and achieves a clear goal

Weaknesses

  1. In places unclear about mathematical assumptions
  2. Should contain a more thorough discussion of existing literature

Report

The goal of this paper is to classify all so called non-regular $4 \times 4$-solutions to the quantum Yang-Baxter equation, using the same techniques as used by one of the authors in [15]. Essentially, under certain analyticity conditions one can set up an inductive algorithm to construct solutions order by order. The authors execute this algorithm, using the known exhaustive list of constant solutions as the seeds that define the lowest order. The wide applicability of such solutions, from quantum computing and condensed matter to potentially high-energy theory, makes that these results have clear potential for follow-up work and are of broad general interest.

The paper is written in an accessible and self-contained way, and tries to prove all non-trivial statements. The overall approach is clear and Section 5, discussing the modified YBE, is interesting.

In places, though, the paper is unclear about the chosen assumptions, and could be improved by more precise formulations, including relevant equations, and a more thorough discussion of existing literature. Specifically, it would be immensely useful for future readers if the authors included an overview of their results, probably in the form of a table (see 11. below).

I recommend the publication of this paper after the minor revisions requested below have been implemented.

Requested changes

  1. The paragraph 'When the local....described by these matrices' in the Introduction introduces what constant solutions are, but after already using the term to describe the reference [6]. More importantly, reference [6] does not study constant solutions, but rather shows that the most general spectral-parameter dependent solution of the YBE is given by the eight-vertex R-matrix. The main assumption in that paper is that the curve on which the spectral parameters live should be elliptic (the non-degenerate case), and more degenerate cases have been studied in subsequent work, e.g. the series of work by Dragovich starting with

V.I. Dragovich, Solutions to the Yang equation with rational spectral curves, Algebra i Analiz 4 (1992), no. 5, 104–116 (Russian)

It would be good to compare the results of this paper with this part of the literature. Also see 11.

  1. The sentences just above (2) says in three different ways that R is assumed to be analytic, which is confusing. Streamlining this would be good.

  2. The assumption discussed under 2. above is somewhat contradicted by the start of Method: in the first paragraph, R is taken to be meromorphic, and then simplified again using a renormalisation argument. It would be good to explain this argument in a little more details, and bring it in line with the discussion under 2.

  3. What is the `basis transformation' that is alluded to in Prop. 1?

  4. It would be clearer to first define what constant solutions are and remark that they have been classified in [7], before stating Prop. 1.
  5. In the last line of the proof, $R(u_+)$ should be $R^{(0)}(u_+)$. Also, the sentence (and the related part of the Proposition) is a bit ambiguous. It states that all constants in [7] must be upgraded to functions, but shouldn't it be: for every $u_+$, $R^{(0)}(u_+)$ should solve the constant YBE, i.e. be one of the solutions from the classification in [7], and so $R^{(0)}(u_+)$ should define a (smooth?) path through this space of solutions.

  6. Above (10): I would appreciate some more details to explain the argument that allows one to set $R^{(0)}$ to $0$.

  7. In Section 2.2, there is no formula for $R^H$ or $R^A$. It would be good to include those here, and otherwise at least provide the equation number in the appendix.

  8. In the first example (and also in all later computations), it is not made very clear how the functional equations that occur are solved, and whether the solutions methods guarantee that all solutions are found. Is the problem really a linear one?

  9. In the second example, in (17) the functions $f_i$ seem to be free, but in a subsequent step they become restricted further. Making clear that (17) is only a first order solution or something like that would aid the reader.

  10. Given that the paper should classify all solutions, I wonder what happens to the solution path given by (19), i.e. what is the non-regular solution that corresponds to that. More generally, it would be of great value to future readers if the paper contained a table with a clear overview of the results, with for each of the constant solutions an overview of the associated non-regular solutions, and ideally whether these are found in [16].

  11. I am puzzled by the start of Section 3: the examples do not look like they have spectral parameters in an overall normalisation, (21) only has spectral parameter dependence in one of its components. Could this statement be clarified?

  12. In line with my comment under 11. it is not very clear where all the results in Section come from, e.g. (25) and (26).

  13. Above (39) it is written that $\log t$ should be well-defined. What is meant by that in this context? The text already states it admits a power series expansion in $u$.

  14. Under (40) the definition of a non-regular Lax matrix is given in the text, and this gets used quite a bit in the subsequent discussion. Giving more prominence to this, new, definition would be helpful.

  15. The argument that simplifies $L_{an}^{-1} R_{an}$ seems to depend on the locality assumptions on $L$ and $R$, i.e. that they only depend on and act on 2 of the tensor legs non-trivially. Making this explicit would be good.

  16. The statement above (77) that all $\tilde{R}$ can be defined such that they have a regular limit seems interesting, but is quite terse. Could the authors expand on that a little bit?

  17. References can be formatted better, in particular [1], [21], [22] (which was published in Journal of Mathematical Sciences) and [23] (in which there is a typo).

  18. There are a number of typos throughout the document, which should be corrected. Here is a small list of some of them:

  19. above (15): solve -> solved
  20. between (20) and (21): solution.. -> solution.
  21. above (41): a an-> an -under (48): indiction -> induction -above (53): relations relations ->relations -In caption of Fig. 1: in a the-> in the -under (78): that -> than -above (79): this if-> this is -last line of Outlook: find the -> to find the

  22. Extra dots in the matrices of 11, 12, 13, 16.

  23. Equations with comma's or dots missing: 11, 12, 20, 63,84, 91, 93, all in appendix B.

Recommendation

Ask for minor revision

---

## Round 3 · Referee Report · Anonymous (Referee 3) · 2025-8-17

Strengths

Addresses an important problem of `classifying' quantum Yang-Baxter equation.

Weaknesses

Classification is a strong term in mathematics and thus requires a proper definition. So the assumptions under which this is a classification has to be clearly spelt out.

Report

Please find my report attached.

Requested changes

Please see attached report.

Attachment

Recommendation

Ask for major revision

---

## Editorial Decision

awaiting_resubmission